# Autoimmune Connective Tissue Diseases-Related Pruritus: Proper Diagnosis and Possible Mechanisms

**DOI:** 10.3390/diagnostics12071772

**Published:** 2022-07-21

**Authors:** Lai-San Wong, Yu-Ta Yen

**Affiliations:** 1Department of Dermatology, Kaohsiung Chang Gung Memorial Hospital and Chang Gung University College of Medicine, Kaohsiung 83301, Taiwan; laisan7@hotmail.com; 2Department of Dermatology, Fooying University Hospital, Pentong 928, Taiwan; 3Institute of Biomedical Sciences, National Sun Yat-Sen University, Kaohsiung 80420, Taiwan

**Keywords:** autoimmune diseases, pruritus, small fiber neuropathy

## Abstract

Pruritus is a well-known bothersome symptom among skin disorders, especially inflammatory skin disorders. Lately, a high prevalence of pruritus in patients with autoimmune connective tissue diseases (ACTDs) has been revealed. Patients with ACTDs may suffer from varying degrees of pruritus, which affect their quality of life. However, it is rarely recognized both by patients and physicians. Meanwhile, pruritus is not only a symptom but is also related to the disease severity of some ACTDs. The pathophysiology of ACTD related pruritus is ambiguous. This review summarizes the features and possible mechanisms of ACTD-related pruritus, which might lead to proper diagnosis and treatment.

## 1. Introduction

Pruritus is a bothersome and common symptom in dermatology. The mechanisms of pruritus are complicated and it emerge from numerous systemic disorders, including dermatologic, neurologic, systemic, psychiatric, and a combination of the above systems. Great advances have been made recently in research into the mechanisms of pruritus and its pathophysiology in inflammatory disorders such as atopic dermatitis and psoriasis [1,2]. However, pruritus in autoimmune connective tissue diseases (ACTDs) is easily neglected and under-recognized by both patients and physicians due to relatively obscure skin changes and unspecific itch patterns. As a matter of fact, pruritus is not rare in patients with ACTDs and 57% of patients with ACTDs complained of pruritus in the initial visit [3]. Importantly, pruritus is sometimes severe and impairs quality of life and mental health [4,5]. In this review, we summarize the prevalence, possible mechanisms (Figure 1), features, and potential treatments in ACTD-related pruritus. This might lead to a proper diagnosis in ACTD patients presenting with pruritus and provide an early diagnosis in certain types of ACTD.

## 2. Dermatomyositis (DM)

### 2.1. Prevalence and Characteristics of Pruritus

DM is an uncommon autoimmune disease involving both the skin and muscle [6]. It is one of the most well-recognized ACTDs presenting with pruritus [7]. The prevalence of pruritus in patients with DM was variable, from more than 50% to higher than 90%, among different studies [8,9,10,11]. The degree of pruritus in patients with DM is often severe and one study of 191 patients demonstrated that more than half of the patients with DM experienced moderate to severe pruritus defined by visual analogue scale (VAS) scores of 3.6–10 [11], which is consistent with previous studies [8,9]. There was no difference in VAS scores between classic DM and clinically amyopathic DM [11]. It has also been found that the degree of pruritus was correlated with the severity of skin disease activities (cutaneous dermatitis area and severity index (CDASI)) and quality of life [9,11]. Importantly, pruritus is the most common initial symptom among the manifestations of DM [10] and it has been reported that DM shows a significantly higher pruritus score than in patients with cutaneous lupus erythematosus (CLE) (VAS scores in DM: 3.8 versus CLE: 2.0) [9]. Therefore, it was suggested that intense pruritus might distinguish clinically similar DM from CLE [9,10].

### 2.2. Mechanisms and Treatments

The pathophysiology of pruritus in DM is unknown. Interleukin (IL)-31, a crucial T-helper-2 (Th2) cytokine participating in the pathogenesis of a variety of pruritic skin disorders [12,13], has been reported to be involved in DM-related pruritus [11]. Enhanced expression of IL-31 and IL-31 receptor A in itchy lesional DM skin compared to nonlesional skin and healthy control skin was revealed [11]. Meanwhile, the level of IL-31 mRNA in lesional skin was correlated with the VAS scores [11]. A reduction of intraepidermal nerve fibers (IENF) without changes in the pepti-dergic nerve fibers was reported in a DM patient who presented with recalcitrant pruritus on the skin of the scalp [14], indicating the possible involvement of small-fiber neuropathy in the pathophysiology of DM-related pruritus. Small-fiber neuropathy caused by damage to the peripheral nervous system of myelinated Aδ-fibers and unmyelinated C-fibers [15] is one of the conditions causing neuropathic itch, which occurs due to injury of neurons of the peripheral or central nervous system [16]. Neuropathic itch has been demonstrated to play a role in a range of itchy disorders [17] and DM-associated sensory neuropathy [18], suggesting that peripheral nerve damage in the inflammatory course of DM may contribute to the mechanism of DM-related pruritus. However, further large studies are required into the association of small-fiber neuropathy and DM-related pruritus.

Accordingly, targeting IL-31 is a potential therapy for DM-related pruritus. Nemolizumab, an IL-31 receptor A inhibitor, has shown promising effects in atopic dermatitis with pruritus in a phase 3 clinical trial [19]. BMS-98116, a humanized IL-31 antibody, and vixarelimab (KPL-716) targeting the OSMR β chain of IL-31 receptor might also be efficacious treatments. Lenabasum (JBT-101, anabasum), a selective cannabinoid receptor type 2 agonist, downregulated the expression of IL-31 from CpG stimulated peripheral blood mononuclear cells in an in vitro study, suggesting a possible role in treatment of DM-related pruritus [11]. Recently, a phase 2 study showed lenabasum improved skin symptoms with a reduction in the scores of CDASI [20]. Though the degree of pruritus was not evaluated in that trial, treatment with lenabasum might be efficacious for DM-related pruritus. A phase 3 trial assessing the efficacy and safety of lenabasum for the treatment of DM is ongoing (NCT03813160). Treatments targeting reduction of the inflammatory process is reasonable in DM-related pruritus. Tacrolimus, a calcineurin inhibitor, has been shown to be effective for the management of pruritus in patients with DM [10]. A patient with recalcitrant cutaneous DM and severe scalp pruritus was responsive to apremilast, an inhibitor of phosphodiesterase 4 specific for cyclic adenosine monophosphate [21]. More clinical trials are necessary for evaluation of the specific roles of the inflammatory components.

## 3. Cutaneous Lupus Erythematosus (CLE)

### 3.1. Prevelance and Characteristics of Pruritus

Pruritus in patients with CLE was considered to be insignificant, especially compared with DM [9]. However, recent studies had inconsistent results. One study of 42 patients reported median pruritus VAS scores were 6 and 4 in patients with specific and non-specific CLE skin lesions, respectively. Though the degree of pruritus was no different between groups of CLE, LE nonspecific lesions, or a combination of both, a correlation between pruritus and CLE disease area and severity index (CLASI) scores was found in the group of specific CLE skin lesions. That study also demonstrated that pain, but not pruritus, was correlated to quality of life [22]. Another multicenter multinational cross-sectional study of 567 patients, which included CLE specific skin lesions, showed that 75% of patients experienced pruritus. Most of the patients (62.1%) with CLE had mild itching (numerical rating scale (NRS) score 1–3). Subjects with acute CLE were most frequently reported to have itch, followed by chronic CLE and subacute CLE. Among the subtypes of chronic CLE, hypertrophic CLE had the highest pruritus intensity (NRS 5.3 ± 3 points) while LE profundus had the lowest pruritus intensity (NRS 1.9 ± 2.7 points). Inconsistent with the aforementioned study, a correlation was revealed between the intensity of pruritus and the severity of skin lesions [23]. Recently, Samotij et al., revealed that 76.8% (116/153) of patients with CLE experienced pruritus and more than half had moderate or severe pruritus (NRS above 4 points). Importantly, they showed that pruritus was not only correlated to skin activities but also systemic LE (SLE) activities [24]. Samotij et al., also found that the most commonly affected itching area was the scalp, followed by the face (excluding the ears and nose) and the arms. The majority of patients experienced pruritus on a daily basis, associated with a burning and tickling sensation [24]. Accordingly, pruritus might have more prominent impacts on patients with CLE than thought. It is necessary to pursue further investigation into the degree of pruritus and the association of pruritus with SLE activities.

### 3.2. Mechanisms and Treatments

The pathogenesis of LE is largely unknown. Small fiber damage and decreased IENF density (IENFD) were found in patients with SLE [25]. Neuropathic itch as an initial presentation in a patient with lupus transverse myelitis has been reported [26]. This indicates that neuropathic itch might participate in the pathogenesis of CLE-related pruritus. Other itch mediators such as IL-6, IL-33, and IL-31 may be involved in the mechanism of CLE-related pruritus. IL-6 was related to the pruritus transduction in prurigo nodularis [27] and calcium phosphate-induced pruritus [28]. Increased expression of IL-6 in the skin of patients with SLE was found [25]. On the other hand, IL-6 can activate IL4+ CD4+ T cells [29], which may indirectly activate downstream itch pathways. Lately, polymorphism of IL-31 [30] and IL-33 [31] was shown to be associated with the risk of SLE. In addition, enhanced serum level of IL-31 in patients with SLE has been reported [32]. IL-33 and IL-31 are pivotal pruritogens in atopic dermatitis, the prototype of inflammatory skin disorder [33]. It is worth specifying the role of these cytokines in the pathogenesis of CLE-related pruritus.

Since little is known about the mechanisms of pruritus in CLE, treatment is mainly based on immunosuppression for LE. Diminishing the inflammation by topical corticosteroid, calcineurin inhibitors, or systemic administration of methotrexate and azathioprine may be considered for CLE-related pruritus. Hydroxychloroquine, one of the mainstay treatments for LE, might induce pruritus [34]. Hydroxychloroquine-induced pruritus is less common and less severe than chloroquine, which induced itch by binding to the Mas-related G protein-coupled receptor (Mrgpr A3/Mrgpr X1) in the periphery and via gastrin-related peptide in the central nervous system [35]. Therefore, it should be taken note of when itching is present in LE patients treated with hydroxychloroquine.

## 4. Systemic Sclerosis (SSc)

### 4.1. Prevalence and Charateristics of Pruritus

SSc is a rare and complex ACTD, which targets connective tissue-producing cells, vascular structures, and the immune system [36]. Skin signs are the earliest and relatively most easily approached manifestations [37]. Itching has been reported as the ninth most frequently experienced and the 31st impacted symptom among 69 symptoms evaluated in a Canadian survey of 856 patients with SSc [38]. It showed that the onset of pruritus was more likely 1.0 to 1.9 years from disease onset [39]. The prevalence of pruritus in patients with SSc varies ranging from more than 40% [40,41], 56.7% [42] and more than 60% [38,43,44] in different studies. This suggests pruritus is a notable feature in patients with SSc. In a questionnaire study of 61 patients, the intensity of pruritus in average was 4.4 ±1.8 with a maximum of 7.1 ± 1.9 measured by VAS scores, indicating the high severity of pruritus in patients with SSc [43]. A substantially stable and persistent itch pattern accompanied by numbness, pain, and burning sensation was described [41,43]. The most commonly affected area was the head and followed by the back and dorsal hands [43]. However, Stull et al., reported that the back was the most common itchy location followed by the arms and scalp [42]. A high impact of pruritus on quality of life with worse mental and physical function, fatigue, and greater disability was found in patients with SSc compared to patients without pruritus [40,45]. In addition to gastrointestinal system involvement and pain, SSc patients with pruritus were significantly associated with sleep disturbance [46]. Intriguingly, Racine et al., showed that the effect of pain severity on sleep and physical function diminished as itch severity increased in patients with SSc [45]. This illustrates the possible interaction between pain and itch pathways [47]. Furthermore, a correlation between pruritus and systemic involvement was reported. A post hoc analysis revealed that SSc patients with pruritus had more skin involvement, gastrointestinal symptoms, worse breathing problems, worse Raynaud’s symptoms, and more severe finger ulcers in a cross-sectional multicenter study of 400 patients [39]. A study of 959 patients from the Canadian Scleroderma Research Group Registry showed that the presence of pruritus was associated with greater skin involvement (*p* = 0.017) and there was a correlation of pruritus between skin severity activities and gastrointestinal system involvement in patients with SSc [41]. However, a cross-sectional survey of 60 patients showed no significant association between pruritus and systemic manifestations of SSc, including interstitial lung disease, pulmonary hypertension, renal involvement, gastrointestinal involvement, or joint pain [42]. More investigations are necessary into the association between pruritus and systemic involvement of SSc.

### 4.2. Mechanisms and Treatments

The pathophysiology of SSc is unclear. Stull et al., reported that there was no association with pruritus and the presence of autoantibodies including anti-nuclear antibodies, anti-double-stranded DNA antibodies, anti-topoisomerase I antibodies, anti-centromere antibodies (ACA), anti-RNA polymerase III antibodies, and anti-U3 RNP [42]. However, Gourier et al., demonstrated that the ACA-positive subgroup had pruritus predominantly in the non-sclerotic areas (82.4%), a longer duration of SSs disease and pruritus, and a higher proportion of patients had preexisted pruritus before the appearance of SSc (17.6% vs. 0%) compared to the ACA-negative subgroup [44]. This indicates a possible immunological component in the mechanism of pruritus in SSc. Rituximab, an anti-CD20 monoclonal antibody, showed improvement in pruritus in 6 out of 8 patients with SSc [48]. In that study, IL-6, IL-15, IL-23, and IL-17 were abnormally increased in three patients and these cytokines were reduced after Rituximab treatment [48]. Further studies are required into the possible role of these cytokines in the pathophysiology of SSc-related pruritus.

There is currently no specified therapy for pruritus in patients with SSc. Since pruritus is significantly associated with xerosis in patients with SSc [42], it is essential to have an emollient as a general principle in daily practice. The British Society for Rheumatology and British Health Professionals in Rheumatology guidelines recommend lanolin-based moisturizer and antihistamine for itch management in patients with SSc [49]. Opioid signaling plays a part in itch perception by modulation of neuronal sensation [50]. Three patients with SSc showed improvement in pruritus and gastrointestinal symptoms with low-dose naltrexone, a mu-opioid receptor antagonist [51]. This indicates the possible involvement of the opioid system in SSc-related pruritus. Enhanced IL-6/JAK/STAT signaling and tofacitinib, a JAK1/3 inhibitor, gene signature were found in skin and lung biopsies from patients with SSc [52]. Meanwhile, tofacitinib improved both the skin and lung fibrosis in a bleomycin-induced scleroderma mouse model [52]. Lately, a pivotal role of the JAK pathway for itch transduction has been revealed [2]. JAK/STAT signaling is responsible for part of the downstream effect of pruritogenic cytokines such as IL-4 and IL-31 [13,53]. Meanwhile, disruption of JAK1 signaling in sensory neurons reduced chronic itch in mice [54]. Ju et al., reported that a significant reduction in itch scores of 3.34 (±3.34) in non-atopic dermatitis chronic itch with 2% topical tofacitinib [55]. The above evidence suggests JAK inhibitors might be a promising treatment for both systemic symptoms and pruritus of SSc.

## 5. Morphea

### 5.1. Prevalance and Characteristics of Pruritus

Morphea, also known as localized scleroderma, is an inflammatory autoimmune disorder characterized by skin and soft tissue sclerosis [56]. Scarce information is available about the prevalence and pattern of pruritus in morphea. Though pruritus was not specifically evaluated, a study of 101 patients showed itch or pain, noted in 46% of patients with morphea [5]. Another study revealed that more than half of patients with morphea (21 of 40, 52.5%) had skin pain, itchiness, tightness, or burning sensations in the affected area and the subjective symptoms were more obvious over the high activity localized lesions [57]. A cross-sectional survey study of 322 patients with morphea showed that itch VAS scores were 3.96 ± 3.17 and 3.42 ± 2.17 in adults and children, respectively [4]. The severity of pruritus in morphea is less intense compared to the other ACTDs mentioned above, but pruritus is one of the most commonly reported symptoms along with pain, numbness, and tingling [5]. Das et al., demonstrated that itching and pain but not fatigue affect the quality of life, along with numbness and tightness, in patients with morphea [4]. Consistent with the above study, Lis-Święty et al., also showed subjective symptoms including itching, pain, tightness, and burning sensation significantly impair health-related quality of life [58]. Additionally, Krof et al., illustrated that itch, pain, and fatigue also impacted on psychological functioning and itch was related to depressed mood in patients with morphea [59]. Generalized morphea had the highest psychological distress compared with linear and plaque types [59]. Moreover, Das et al., revealed a significant correlation between pruritus and lesion activity [4].

### 5.2. Mechanisms and Treatments

There are limited studies on the pathophysiology of pruritus in morphea. Therefore treatment mainly targets anti-inflammation for morphea itself. Phototherapy is one of the most well-studied therapies in treatment of morphea [60]. A case series revealed initial response for itching measured by VAS scores with narrowband ultraviolet B (UVB) at 5 to 9 sessions in different types of morphea [61]. However, only medium-dose UVA1 demonstrated significantly improved itching compared with low-dose UVA1 and narrowband UVB phototherapy in a randomized control study [62]. The anti-pruritic effect of phototherapy is complicated. UV-light might modulate the cutaneous sensory nervous system for pruriception directly or function on skin immune cells indirectly, which further reduces the release of pruritogenic mediators [63]. For the possible role of inflammatory components in the pathogenesis of morphea-related pruritus, less itchiness was reported with abatacept, an inhibitor of T cell activation, in a patient with morphea profunda [64].

## 6. Sjögren Syndrome (SS)

### 6.1. Prevalence and Characteristics of Pruritus

SS is an autoimmune disease characterized by dysfunction of salivary and lacrimal glands. It can be defined as primary SS when it occurs alone or secondary SS if systemic autoimmune diseases accompany [65]. Extra-glandular involvement is common, especially dermatological involvement. An Italian retrospective study of 93 patients showed that pruritus was found in 41.6% and 38.3% of primary and secondary SS, respectively [66]. In that study, pruritus was the second most frequent cutaneous manifestation following xerosis, which was more frequently noted in primary SS than secondary SS (56.4% versus 25.8%) [66]. Another study of 19 patients showed that 53% of patients with primary SS experienced pruritus and the mean duration of pruritus was 74.4 months. The intensity of pruritus was 7.7 ± 1.7 measured by VAS scores and the severity of pruritus had a significant correlation with quality of life. The most affected location was the shins, followed by the back and forearms [67]. Furthermore, xerosis was the most common aggravating factor for pruritus in primary SS and 90% of pruritic patients also suffered from xerosis while only 44% did in the non-itchy group [67], indicating the influence of xerosis on itchy sensation.

### 6.2. Mechanisms and Treatments

The mechanisms of pruritus in SS remain elusive and might be multifactorial. Xerosis, the most common symptom in SS, might contribute to or at least aggravate pruritus in SS. Nevertheless, the pathogenesis of xerosis in SS is controversial. It is suggested that xerosis in SS was caused by dysfunction of sweat glands by lymphocytes-mediated injury [68], but Bernacchi et al., reported that there was no sweat glands impairment, but there were biochemical alterations of the epidermis with increased epidermal proliferation and perturbation of epidermal differentiation in patients with SS [69]. Further investigations of the possible mechanisms of xerosis and the relationship between xerosis and pruritus in SS are necessary. Peripheral nervous system involvement with sensory neuropathy in SS is common [65,70]. Small-fiber neuropathy has been demonstrated in primary SS and higher neuropathic symptom burden compared to other well-known small-fiber neuropathy etiologies, hereditary transthyretin amyloidosis and idiopathic small-fiber neuropathy [71]. Therefore neuropathic itch may be involved in pruritus of patients with SS [17]. Due to the unclear pathogenesis of pruritus in SS, specialized pruritus therapy is sparse. Moisturizer as general skin care, to avoid xerosis and additional scratching, is required. For the possible role of neuropathic itch in the pathogenesis of SS, neuro-modulating agents, such as gabapentin and pregabalin might be considered [72].

Of note, a meta-analysis of 14 studies reported an increased risk of malignancies, non-Hodgkin lymphoma, and thyroid cancer in patients with primary SS [73]. Another large cohort study of 1300 patients revealed B cell lymphoma is the most frequently associated cancer with primary SS [74]. Paraneoplastic pruritus, especially lymphoma and leukemia, is one of the important etiologies of idiopathic generalized pruritus [75]. Although pruritus is less common in non-Hodgkin lymphoma than in Hodgkin lymphoma, 15% of patients with non-Hodgkin lymphoma experienced generalized itching [76]. A detailed survey of occult neoplasia, especially hematological malignancies, is required when unexplainable pruritus occurs in patients with stable disease activity of SS.

## 7. Conclusions

Pruritus is a frequent presentation in ACTDs and physicians should pay attention to this prominent symptom. Comprehensive medical history and physical examination are essential to disclose the characteristic signs and symptoms of pruritus in patients with ACTDs. A good understanding of the features and pattern of pruritus in ACTDs may avoid delayed diagnosis and misleading diagnosis of inflammatory or allergic diseases. Pruritus in some types of ACTDs not only affects quality of life but is also associated with skin disease activities, such as DM and CLE. Current evidence provides more insights into the possible pathophysiology and therapies in various types of ACTDs (Table 1), but there is still a knowledge gap regarding a more specific molecular pathway in ACTDs-related pruritus. Extensive studies into the different features of pruritus in various subtypes of each ACTD are also important. As the mechanisms of ACTD-related pruritus are generally unknown, specialized treatment is insufficient. More investigations are warranted to develop advance targeting therapies for this distressing symptom in ACTDs.

## Figures and Tables

**Figure 1 diagnostics-12-01772-f001:**
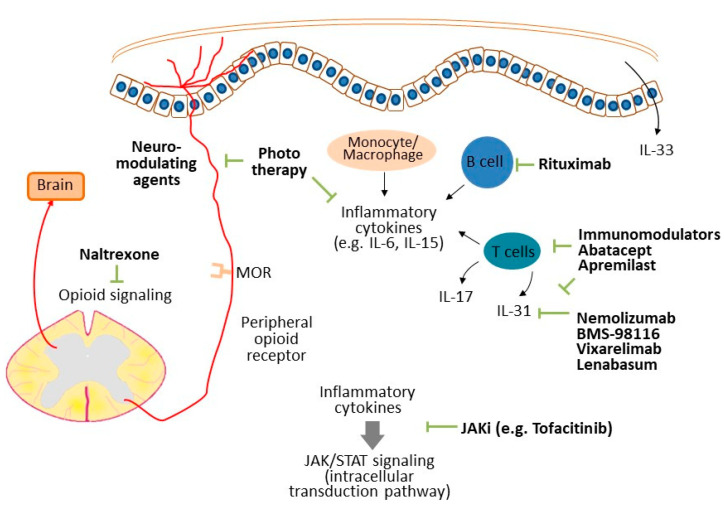
The possible mechanisms of ACTD-related pruritus. Inflammatory cytokines, such as IL-33 and IL-31, released from keratinocytes and immune cells transduce the itch sensation directly. Meanwhile, other inflammatory cytokines might participate in pruritus transduction through activation of the JAK/STAT pathway. Small-fiber neuropathy and opioid signaling also modulate the perception of pruriception. Therapies targeting the above mediators might reduce the intensity of itch in patient with ACTDs.

**Table 1 diagnostics-12-01772-t001:** Summary of studies of possible mechanisms for ACTD-related pruritus and potential targeted treatments.

Proposed Mechanisms	Potential Targeted Treatments
**Dermatomyositis**	
IL-31 [11]	Nemolizumab, BMS-98116, vixarelimab, lenabasum
Small-fiber neuropathy [14]	Neuro-modulating agents (e.g., gabapentin, pregabalin)
Inflammatory components	Tacrolimus [10], apremilast [21]
**Cutaneous lupus erythematosus**	
Small-fiber neuropathy [25]	Neuro-modulating agents (e.g., gabapentin, pregabalin)
Neuropathic itch [26]	Neuro-modulating agents (e.g., gabapentin, pregabalin)
IL-6 [25], IL-33 [30], IL-31 [31]	Topical corticosteroid, calcineurin inhibitor, methotrexate, azathioprine
**Systemic sclerosis**	
Xerosis [42]	Emollient [49]
IL-6, IL-15, IL-23, IL-17 [48]	Rituximab [48]
Opioid signaling	Naltrexone [51]
JAK/STAT signaling [52]	JAK inhibitor (e.g., tofacitinib) [52]
**Morphea**	
Inflammatory components	Phototherapy [61,62], abatacept [64]
**Sjögren syndrome**	
Xerosis [67]	Emollient
Small-fiber neuropathy [71]	Neuro-modulating agents (e.g., gabapentin, pregabalin)

The proposed mechanisms of different ACTDs and the possible therapies were summarized. IL, interleukin; JAK/STAT, Janus kinase/signal transducer and activator of transcription. The data above were retrieved before 30 June 2022.

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
