# Peer review of "Autoimmune Connective Tissue Diseases-Related Pruritus: Proper Diagnosis and Possible Mechanisms"

_diagnostics, 2022, doi:10.3390/diagnostics12071772_

Round 1
Reviewer 1 Report
1. It is strongly recommended to change the title from “Recognition of autoimmune connective tissue diseases-related 2 pruritus: proper diagnosis and possible mechanisms” to “Autoimmune connective tissue diseases-related pruritus: proper diagnosis and possible mechanisms”.
2. Keywords should be revised and expanded using MeSH.
3. This manuscript is a “narrative review”. Therefore, the section of “” is not required. However, the contents under this title should be considered as your results.
4. For each of the five discussed disease, all of the contents are under a subheading. They should be rearranged. For instance, a subheading should be assigned for prevalence and intensity of pruritus in each of the disease. Another one for mechanism, etc.
5. A figure for presenting the main mechanisms is necessary. It should be added for better and easier understanding about this important point.
6. In the conclusion, some discussions should be added.
7. This sentence seems to be irrelevant and should be omitted “6. Patents This section is not mandatory but may be added if there are patents resulting from the work reported in this manuscript.”.
Author Response
-
- It is strongly recommended to change the title from “Recognition of autoimmune connective tissue diseases-related 2 pruritus: proper diagnosis and possible mechanisms” to “Autoimmune connective tissue diseases-related pruritus: proper diagnosis and possible mechanisms”.
We appreciate the valuable suggestion and have changed the title to “Autoimmune connective tissue diseases-related pruritus: proper diagnosis and possible mechanisms”.
- Keywords should be revised and expanded using MeSH.
Thank you for the suggestion and we have revised and expanded using MeSH. The revised version is “Autoimmune diseases; pruritus; small fiber neuropathy”
- This manuscript is a “narrative review”. Therefore, the section of “” is not required. However, the contents under this title should be considered as your results.
Thank you for the comment. It seems that the “Material and method” was a wrong input by the system. Therefore, we have already omitted “Material and methods” but kept the section.
- For each of the five discussed disease, all of the contents are under a subheading. They should be rearranged. For instance, a subheading should be assigned for prevalence and intensity of pruritus in each of the disease. Another one for mechanism, etc.
We appreciate the suggestion and have added the subheading as “Prevalence and characteristics of pruritus” and “Mechanisms and treatments” in each section.
- A figure for presenting the main mechanisms is necessary. It should be added for better and easier understanding about this important point.
We appreciate the valuable suggestion. Therefore we have added table 1 to the manuscript.
- In the conclusion, some discussions should be added.
Thank you for the comment. We have added more discussion in the last section “Current evidence provides more insights into the possible pathophysiology in various types of ACTDs, however, there is still a knowledge gap on a more specific molecular pathway in ACTDs-related pruritus. Extensive studies such as the different features of pruritus in various subtypes of each ACTD are important as well.”
- This sentence seems to be irrelevant and should be omitted “6. Patents This section is not mandatory but may be added if there are patents resulting from the work reported in this manuscript.”.
Thank you for the comment. It seems that this sentence was a wrong input by the system. Therefore, we have already omitted it.

Reviewer 2 Report
This review is rather preliminary for publication and includes a lot of problems as follows:
1.Any scheme showing the mechanisms for the pruritus in ACTDs is necessary.
2.' Neuropathic itch' is often used in the text. Please explain the term in detail when the term is firstly used.
3.Line 36:2.Materials and methods
This title is rather queer, and inappropriate, and should be corrected.
4.CLE includes so many different types of eruption, DLE, acute CLE, SCLE, chilblain lupus, LE tumidus, etc. So it is not recommended to use totally CLE. Please explain the pruritus in each type of CLE.
5.Line 129: Since chloroquine receptors exist on sensory nerves, and mediate chloroquine-induced itch, please explain that mechanism.
6.Lines 187-8: does tofacitinib reduce the pruritus or scratching behavior in scleroderma model mice?
7.Regarding systemic sclerosis, is the intensity or area of sclerosis correlated with the intensity of pruritus?
7.Several grammatical errors should be corrected.
L211 were is not needed; L212 itchy must be itch.
Author Response
This review is rather preliminary for publication and includes a lot of problems as follows:
- Any scheme showing the mechanisms for the pruritus in ACTDs is necessary.
We appreciate the valuable suggestion. Therefore we have added table 1 to the manuscript.
- ' Neuropathic itch' is often used in the text. Please explain the term in detail when the term is firstly used.
Thank you for the comment. We have added the definition of neuropathic itch in the manuscript “Small-fiber neuropathy caused by damage to the peripheral nervous system of myelinated Aδ-fibers and unmyelinated C-fibers [15] is one of the conditions causing neuropathic itch, which occurs due to injury of neurons of the peripheral or central nervous system [16].”
- Line 36:2.Materials and methods. This title is rather queer, and inappropriate, and should be corrected.
Thank you for the comment. It seems that the “Material and method” was a wrong input by the system. We have already omitted “Material and methods” but kept the section.
- CLE includes so many different types of eruption, DLE, acute CLE, SCLE, chilblain lupus, LE tumidus, etc. So it is not recommended to use totally CLE. Please explain the pruritus in each type of CLE.
We appreciate the suggestion. We agree with the importance of the different features in CLE subtypes, however, there are limited studies about the features of pruritus in various subtypes of CLE. The latest three studies (References 22-24), which we included in our review, classified the lesions as CLE specific and non-specific. There was only one study, which was conducted by Samotij et al. (Reference 24), showed the different intensity of pruritus in various subtypes of CLE. Therefore, we used CLE instead of various subtypes. We have added more discussion about the current data on the different features of itch intensity in various subtypes of LE “Subjects with acute CLE were most frequently reported to have itch, followed by chronic CLE and subacute CLE. Among the subtypes of chronic CLE, hypertrophic CLE had the highest pruritus intensity (NRS 5.3±3 points) while LE profundus had the lowest pruritus intensity (NRS 1.9±2.7 points).”. On the other hand, it indicated further investigation is necessary for the subtypes of CLE and we added more discussion in the last section.
- Line 129: Since chloroquine receptors exist on sensory nerves, and mediate chloroquine-induced itch, please explain that mechanism.
Thank you for the comment. We have added the mechanism of chloroquine-induced itch in the manuscript “Hydroxychloroquine-induced pruritus is less common and less severe than chloroquine, which induced itch by binding to Mas-related G protein- coupled receptor (Mrgpr A3/Mrgpr X1) in the the periphery and via gastrin-related peptide in the central nervous system [34]”.
- Lines 187-8: does tofacitinib reduce the pruritus or scratching behavior in scleroderma model mice?
Thank you for the question. There was no evaluation of the influence of tofacitinib on pruritus or scratching behavior in the scleroderma mouse model in that study.
- Regarding systemic sclerosis, is the intensity or area of sclerosis correlated with the intensity of pruritus?
Thank you for the question. Razykov et al. showed “The presence of pruritus was independently associated with greater skin involvement [odds ratio (OR)=1.02, 95% CI 1.00, 1.04, P=0.017] ” and we have added this information in our manuscript “A study of 959 patients from the Canadian Scleroderma Research Group Registry showed the presence of pruritus was associated with the greater skin involvement and…”.
- Several grammatical errors should be corrected.
L211 were is not needed; L212 itchy must be itch.
We appreciate the suggestion. We have rechecked the manuscript carefully and corrected the mistakes as you can see in the new version. We also corrected the grammatical errors and also omitted “were” and corrected“itchy” to “itch” as you suggested.

Round 2
Reviewer 2 Report
Though the authors add the possible mechanism and target treatment, that is not what is required. The figure of scheme which showing each disease skin lesion al status of immune cells , keratinocytes, fibroblasts, and cytokines/chemokines and the status of sensory neurons, and the target points of some drugs such as antiIL-31 antibody.
In other aspects, the authors well addressed the issues I previously pointed out. Thus the figures I mentioned above is the necessity for the acceptance of publications.
Author Response
We appreciate the valuable suggestion and have added figure1 according to the possible mechanisms in ACTDs-related pruritus.
Round 3
Reviewer 2 Report
The manuscript is improved by the addition of figure. This manuscript is now acceptable for publication